# Active forest stewardship benefits priority birds in the New Jersey Pine Barrens

Christopher K. Williams[ID][1]*, Theron M. Terhune, II[2¤a], John Parke[3¤b], Elizabeth A. Matseur[3], John Cecil[3¤c]

**1** Department of Entomology and Wildlife Ecology, University of Delaware, Newark, Delaware, United States of America, **2** Tall Timbers Research Station, Tallahassee, Florida, United States of America, **3** New Jersey Audubon, Port Murray, New Jersey, United States of America

¤a Current address: Orton Plantation, Winnabow, North Carolina, United States of America
¤b Current address: Merrill Creek Reservoir, Washington, New Jersey, United States of America
¤c Current address: New Jersey Department of Environmental Protection, Trenton, New Jersey, United States of America
* ckwillia@udel.edu

**Data Availability Statement:** The data is now stored at Harvard Dataverse, 'Pine Barrens Songbird Data', https://doi.org/10.7910/DVN/RDOJ4M.

## Abstract

Fire suppression has negatively impacted thousands of acres of private and public lands in the United States. As a case study, the New Jersey Pine Barrens (NJPB) are a disturbance driven ecosystem that is experiencing serious ecological implications due to a loss of traditional forest thinning activities such as harvesting for forest products or thinning for wildfire fuel-load reduction measures coupled with a long-standing philosophy of fire suppression and dormant-season prescribed burning. Dense closed-canopy forest conditions, dissimilar to historic open-canopy forests of the NJPB, have reduced abundance and diversity of certain flora and fauna, including regionally imperiled breeding birds. In recent years, active forest stewardship (e.g., thinning, clear-cutting, and burning) has occurred on private and some public lands within the NJPB; however, the impact of such management on breeding birds is unclear due to a paucity of research on this subject within the NJPB. During 2012, 2013, 2016, and 2017, we conducted repeat-visit point counts ($n = 1,800$) for breeding songbirds across 75 control and 75 treatment sites within the NJPB to assess the influence of forest structure at three strata levels (groundcover, midstory profile, and canopy) on breeding bird communities. Specifically, we constructed a hierarchical community abundance model within a Bayesian framework for Bird Conservation Region (BCR) 30 priority upland birds ($n = 12$) within three species suites: Forested Upland, Scrub-Shrub (or Young Forest), and Grassland. At the community level, we found a negative relationship between bird abundance and live tree basal area. At the BCR 30 suite level, we found no relationship between Forested Upland suite-level abundance and any of the measured covariates; however, we found a negative relationship between percentage of woody groundcover and Scrub-Shrub suite-level abundance, and negative relationship between horizontal visual obstruction at 2 m above ground level and Grassland suite-level abundance. Furthermore, the two latter species suites exhibited a strong negative relationship with basal area. We recommend active forest stewardship that specifically targets opening the canopy to achieve basal areas between ~0–15 m²/ha via selective thinning, shelter cutting, and small-scale clear cutting. Mechanical treatment and prescribed burning would produce such conditions and have the

**Funding:** New Jersey Audubon Society (05012015) served as the primary sponsor and assisted with study design and preparation of the manuscript. Secondary funding came from USDA Hatch (DEL00774) and the University of Delaware Waterfowl and Upland Gamebird Center and had no role in study design, data collection and analysis, decision to publish, or preparation of the manuscript.

**Competing interests:** NO authors have competing interests

added benefit of reducing fuel loads across this ~4,500 km$^2$ landscape as well as assisting in carbon defense strategies for the region.

## Introduction

Birds are often used as bioindicators of forest quality and management [1–6]. While often contiguous mature forests improve biodiversity, other fire dependent ecosystems rely on thinning to support a more diverse community with increased seed and insect abundance [7–11]. Fire suppression has negatively impacted thousands of acres of private and public lands in the United States [12]. Atlantic Coastal Plain pine-dominated ecosystems evolved with fire-tolerance and fire-dependence [13, 14]. However, fire suppression in the eastern United States has decreased fire-mediated land area [15, 16]. Specifically, pitch pine-scrub oak forests in the Atlantic Coastal Plain that are not burned or harvested have fundamentally changed in species composition and structure, in some cases resulting in dense basal areas, closed canopies, and overgrown woody understories [17–22]. These are undesirable forest conditions for dependent wildlife species and wildfire risk mitigators [23].

The New Jersey Pine Barrens (NJPB) are among the largest remaining segments of pitch pine-scrub oak forest within the Mid-Atlantic United States [24, 25]. Throughout the last 300 years, NJPB forests were harvested several times over to provide fuel and raw materials for a series of industries such as iron forges, charcoal making, glassblowing, and shipbuilding [26]. However, by the beginning of the 21$^{st}$ century, timber harvest had dwindled dramatically. In 1978, the NJPB was classified as a United States Biosphere Reserve and includes portions of seven southern New Jersey counties, and encompasses over 400,000 ha of farms, forests, and wetlands. The reserve supports dozens of rare plant and animal species and occupies 22% of New Jersey's land area, making it the largest body of open space on the Mid-Atlantic seaboard between Virginia and Massachusetts [26, 27].

Currently, the NJPB has widespread dense closed canopy mature pine stands (>20 m$^2$/ha) due to dormant season burning [23], the decline in active timber harvest, and the suppression of natural fire cycle which has largely been controlled to protect the increasing number of people and structures found throughout the Pinelands [26]. As a result, abundance and diversity of priority breeding birds has decreased, with extirpation of some previously common species dependent on open forest and/or early successional conditions (e.g., northern bobwhite and ruffed grouse).

The current management paradigm of the NJPB is to improve forest conditions in a manner that reduces wildfire fuel loads, promotes carbon defense strategies (that protect the integrity of the existing carbon pool rather than trying to expand the pool [26]), and positively impacts both native flora and fauna. While mammal communities have been studied in the context of forest disturbance within the NJPB [which showed greater diversity with disturbance, 28], there is no such research on the response of breeding bird communities to forest management. Among the highest priority objectives of the U.S. Fish and Wildlife Service's Atlantic Coast Joint Venture and Mid-Atlantic and New England Coastal Bird Conservation Region 30 Implementation Plan was to "determine how to manage lands to best achieve (priority bird) species population targets, while minimizing inter-species conflicts" including 12 upland bird species that were of "high" and "highest" priorities for conservation [29] that had local home ranges during the breeding season as compared to nomadic species (e.g. raptors). Further, we followed the Joint Venture lumping of those 12 species into 3 habitat suite-level

types including Forested Upland, Scrub-Shrub/Early Successional/Young Forest (hereafter Scrub-Scrub), and Grassland to assure analyses were not biased due to differences in habitat structure. These included six Forested Upland species (Baltimore oriole [*Icterus galbula*, BAOR], black-and-white warbler [*Mniotilta varia*, BAWW], great crested flycatcher [*Myiarchus crinitus*, GCFL], northern flicker [*Colaptes auratus*, NOFL], scarlet tanager [*Piranga olivacea*, SCTA], and wood thrush [*Hylocichla mustelina*, WOTH]), five Scrub-Shrub species (blue-winged warbler [*Vermivora cyanoptera*, BWWA], brown thrasher [*Toxostoma rufum*, BRTH], eastern towhee [*Pipilo erythrophthalmus*, EATO], field sparrow [*Spizella pusilla*, FISP], and prairie warbler [*Setophaga discolor*, PRAW]), and one Grassland species (eastern kingbird [*Tyrannus tyrannus*, EAKI]). While general management principles may be known for these species, the quantitative impact of active forest stewardship practices within the NJPB on community-level (Pine Barrens), suite-level (Forested Upland, Scrub-Shrub, and Grassland) and species-level abundance is yet unknown.

Our objective was to evaluate the influence of forest structure on the 12 NJPB high priority breeding bird species abundances. Specifically, we aimed to address the influence of active forest stewardship practices (e.g. timber thinning and periodic burns), versus unmanaged lands (i.e., no active management or disturbance), that altered groundcover composition, midstory structure, and canopy on community-level, suite-level, and species-level breeding bird abundance and diversity. We hypothesized that Grassland and Scrub-Shrub species suite abundances would be greater at sites where active forest stewardship had been implemented to open the canopy compared to areas where no management occurred under a closed canopy. Additionally, we hypothesized interior Forested Upland nesting species that are sensitive to edge effects would be negatively impacted by active forest stewardship management practices. Like research exploring forest management on avian biodiversity and abundance in other parts of the United States [30, 31], our goals were to inform conservation planning decisions for the BCR30 and NJPB.

## Study area

The NJPB lie within the Atlantic Coastal Plain physiographic region and span ~445,000 ha across seven counties in southern New Jersey. The landscape is comprised of several large contiguous public forests, including Wharton State Forest (49,728 ha), Brendan T. Byrne State Forest (15,071 ha), Bass River State Forest (11,795 ha), Penn State Forest (1,362 ha), Franklin Parker Preserve (3,804 ha), and Greenwood Wildlife Management Area (13,010 ha). The topography is principally low relief, gently rolling hills composed predominately of acidic sandy soils [32]. The mean temperature in southern New Jersey ranges from 0.17°C in January to 24.28°C in July, with an average annual precipitation of 114.43 cm [33].

The region is characterized by upland pine-oak (predominantly *Pinus rigida* and *P. echinata*) forests, with Atlantic white cedar (*Chamaecyparis thyoides*) swamps, bogs, and graminoid marshes in lowland areas [25]. Less common canopy species within uplands include black oak (*Q. velutina*), white oak (*Q. alba*), chestnut oak (*Q. montana*), post oak (*Q. stellata*), and scarlet oak (*Q. coccinea*). The understory is generally composed of scrub oaks, including dwarf oak (*Q. prinoides*), scrub (bear) oak (*Q. ilicifolia*), and chinquapin oak (*Q. muehlenbergii*); ericaceous shrubs, including northern highbush blueberry (*Vaccinium corymbosum*), lowbush blueberry (*V. pallidum*), and black huckleberry (*Gaylussacia baccata*); and Pine Barren golden heather (*Hudsonia ericoides*). In areas where fire or mechanical treatment has been implemented, native herbaceous groundcover, including little bluestem (*Schizachyrium scoparium*), switchgrass (*Panicum virgatum*), broomsedge (*Andropogon virginicus*), and bracken fern (*Pteridium aquilinum*) were established along with native forbs, including tick-trefoils

(*Desmodium* spp.) and bush-clover spp. (*Lespedeza spp.*). Some open areas contain bare substrate (coastal plain sands, [32]), as well as patches of juniper moss (*Polytrichum juniperinum)* and lichens (predominantly *Cladonia* spp.).

## Methods

### Point counts

We conducted 100 m single-observer fixed-radii circular-plot point count surveys (hereafter "point count") thrice each year from 16 May to 10 August 2012, 2013, 2016 and 2017 at 150 unique survey points across 18 survey sites within the NJPB study area. Half of these sites represented managed upland (hereafter "treatment") sites, with the other half representing unmanaged reference (hereafter "control") sites. Treatment and control sites were used to incorporate a broad range of site characteristics, specifically targeting vegetation structure and composition at three strata levels: groundcover, midstory, and canopy. Treatment sites, which were on both private and public lands, did not receive uniform treatment, but rather were a mixture of management practices that incorporated thinning, clear-cutting, roller chopping, and prescribed burning. Each site was $\geq$1000 m away from the nearest adjacent site and points within a site were spaced $\geq$200 m apart and placed on or adjacent to non-primary roads. We conducted 10-min point counts [34–36] and we recorded all individual birds for each species detected and binned them into two distance categories: 0–50 m or 50–100 m. Unique individuals of the same species were distinguished via simultaneous (or near simultaneous) auditory or visual detection, as well as distinctive voice or plumage, minimizing double-counting. We started surveys 15 min before local sunrise and ended no later than 3.5 hours after. Observers recorded five detection covariates at each point: temperature, humidity, and wind speed using a Kestrel 3000 Handheld Weather Meter (Nielsen-Kellerman Co., Boothwyn, PA, USA), sky condition (cloud cover; scale = 0–4), and disturbance (ambient noise from traffic or aircraft; scale = 0–4). Within each year, we surveyed sites in a randomized order to account for variable detection probability with time of day [37] and every point was surveyed at ~3-week intervals, avoiding inclement weather events (e.g., wind speed $\geq$15 km/h and high precipitation), until three visits were completed. Due to staff turnover, three observers were used over the four years of the study, each trained in identical methodology.

### Vegetation measurements

We measured ten vegetation covariates to capture groundcover, midstory, and canopy components at each survey point once during June–July 2012–13 and 2016–2017. We assumed here that environmental covariates were measured at the appropriate time to capture habitat selection [38]. We recorded groundcover (relative percentages of bare substrate, grass, forbs, leaf litter, and live/dead woody vegetation <2 m; components summed to 100%) within a 4-m$^2$ sampling frame [39] centered on the survey point. We used a three-section modified Nudds board [40] to estimate the vegetation profile via a visual obstruction reading (VOR) of the midstory at three strata levels (ground level [0.25 m; VOR025], 1 m [VOR1], and 2 m [VOR2]). We placed the Nudds board 10 m from the sampling point in each of the four cardinal directions and viewed horizontally from the center of the sampling frame. We considered a square covered if any part of it was obstructed by vegetation. We estimated basal area using a Jim-Gem factor 10 prism (Forestry Suppliers, Inc., Jackson, MS, USA) and converted to m$^2$/ ha. We measured percent canopy closure as the mean of four readings taken in each of the cardinal directions using a convex spherical densitometer at each sampling point (Forestry Suppliers, Inc., Jackson, MS, USA). In the case where a survey point was on a road, we sampled immediately adjacent to the road edge. These groundcover, midstory profile, and canopy

estimates constituted the first replicate for one survey point. The entire sampling frame was replicated twice more at locations 25 m away from the initial sampling frame along a randomly directed azimuth. The mean of the three replicates for each covariate was used to define each location. We assumed that basal area (m²/ha) and groundcover composition remained relatively constant within a survey season. Additionally, while visual obstruction may change as natural vertical growth accelerates during the growing season, we assume this was relatively unchanged in our study area. We made this assumption because of 1) slow vertical growth associated with the highly xeric and "barren" landscape [25], and 2) ancillary data (P. Coppola, University of Delaware, *unpublished data*) during which vegetation measurements at specific points were consistent throughout the 2016–2017 survey periods.

## Statistical analysis

We used a hierarchical community abundance (*N*-mixture) mixed effects model without data augmentation accommodating multiple-visit point count data to account for imperfect detection [41–43]. We assumed counts of species *i* at point *j* in year *r* depended on the latent abundance $M_{ijr}$ following the general process model:

$$a_{ijr} \sim \text{Bernoulli}(\phi_{it})$$

$$M_{ijr} | a_i \sim \text{Poisson}(a_{ijr}\lambda_{ijr}),$$

where $a_{ijr}$ was a species- (*i*) and site-specific (*t*) zero-inflation parameter, given overdispersion of count data. We modeled $\lambda_{ijr}$ as a function of both fixed and random effects:

$$\log(\lambda_{ijr}) \sim \beta_{0(i)} + \beta_{cov(i)} \times \boldsymbol{x}_{(jr)} + \gamma t$$

$$\beta_{0(i)} \sim \text{Normal}(\mu_{\beta 0}, \sigma^2_{\beta 0})$$

$$\beta_{cov(i)} \sim \text{Normal}(\mu_{\beta cov}, \sigma^2_{\beta cov})$$

where $\beta_{0(i)}$ is the intercept, $\beta_{cov(i)}$ is the vector of regression parameters, $\boldsymbol{x}_{(j,r)}$ is the vector of site vegetation covariates, and $\gamma_t$ is the random effect of site *t*. We decided to not include a random effect of year in final models due to observed low significance in early models and confounding model performance with measured vegetation covariates, which was the focus of the study objectives. This was an implicit consequence of the space-for-time replacement experimental design. The second hierarchical level, accounting for imperfect detection, assumed that the number of individuals exposed to sampling (present within sampling radius and signaling) during visit *v* ($N_{ijrv}$) follows a binomial process:

$$N_{ijrv} \sim \text{Binomial}(M_{ijr}, \varphi_{ijrv}),$$

where the detection rate $\varphi_{ijrv}$ is a composite of presence probability, signaling probability, and detectability [43, 44]. We modeled $\varphi_{ijrv}$ as a function of visit-specific covariates:

$$\text{logit}(\varphi_{ijrv}) \sim \alpha_{0(i)} + \alpha_{cov(i)} \times \boldsymbol{v}_{(jrv)}$$

$$\alpha_{0(i)} \sim \text{Normal}(\mu_{\alpha 0}, \sigma^2_{\alpha 0})$$

$$\alpha_{cov(i)} \sim \text{Normal}(\mu_{\alpha cov}, \sigma^2_{\alpha cov})$$

where $\alpha_{0(i)}$ is the intercept, $\alpha_{cov(i)}$ is the vector of regression parameters, and $\boldsymbol{v_{(jrv)}}$ is the vector of visit covariates (e.g., date, wind, disturbance, etc.). We did not include observer effects because it would have been confounded with year, given surveyors differed between years with no overlap. We estimated derived parameters, detection probability ($P_d$) and zero-inflation ($\phi$), as well as the presence/absence matrix ($z$-matrix).

The hierarchical model accounted for species-specific detection probabilities while incorporating visit-level covariates (e.g., date, wind, disturbance) on detection; thus, one hierarchical level is implied for the observation process. Implicitly modeling detection probability as a composite of presence probability ($p_p$; temporary spatial emigration), availability probability ($p_a$; random temporary emigration), and detectability ($p_d$; *sensu* [43]), rather than explicitly modeling these two additional hierarchical processes, comes with a few assumptions [42, 45]. The first assumption is that occupancy is constant across visits (i.e., $p_p = 1.0$); however, this may be violated due to temporary spatial emigration [42]. Temporary spatial emigration is the process by which individuals physically move outside the maximum point count radius, due to incomplete overlap of point count radii and individual home range/territory, thus making them periodically unavailable to observation [42]. If home range sizes vary greatly among species, then this assumption may introduce bias in estimates [46]. Accordingly, we restricted our modeling effort to the 12 priority species which did not include nomadic and aerial coursing species (e.g., raptors), which may have home ranges that are orders of magnitude larger than those species targeted here. Additionally, we did not construct rarefaction curves to calculate density, which would have caused overestimation bias due to a mismatch between the sampling radius and the effective sampling area [41, 43]. Therefore, while home ranges (or territory sizes) do exhibit inter-, as well as intra-, specific variation among the species studied here, that variation is treated as negligible in the context of $p_p$ at point count locations. The second major assumption is that the availability of a bird that is present within the maximum point count radius is constant (i.e., $p_a = 1$). This issue may arise because of an observer causing birds to remain still and silent during the initial moments of a point count. The most common means of dealing with this issue is the use of time-of-detection or time-removal models [44, 47]. We accounted for this issue via a 1-min acclimation period before each point count, allowing ambient calling to return to normal levels. Overall, the detection probability estimates reported from the model are greater than those typically recommended [48] and additional hierarchical levels are not essential in accounting for observational process [44].

We used the $z$-matrix to estimate beta diversity via the Jaccard index ($J$), which expresses the similarity between two species in terms of the points where they co-occur [42, 49, 50]. Values of $J$ range from 0 to 1, corresponding respectively to the extremes of no co-occurrence and perfect co-occurrence. We interpreted $J$ as the similarity between two species in terms of the sites where they both occur, and where they occur alone by:

$$J_{i,j} = \frac{\sum z_i z_j}{\sum z_i + \sum z_j - \sum z_i z_j}$$

where $z_i$ and $z_j$ are the $z$-matrices for species $i$ and $j$, respectively [42, 50]. We computed all pair-wise comparisons between all ($n = 12$) species, thus propagating uncertainty in estimates through the community abundance model [51].

We standardized all continuous covariates, centering each on a mean of 0 and standard deviation of 1, to improve model convergence and interpretability. We used Pearson's correlation tests to assess the degree of collinearity of model parameters and did not fit models when $|r| > 0.7$. We specified vague, normal priors with a mean of zero and precision of 0.001 for all fixed effects [42, 46, 52]. We ran all models using the R2JAGS package [53] in R version 3.6

[54], estimating posterior distributions using MCMC methods with 3 independent chains. Each chain ran 200,000 iterations, discarding the first 100,000 and saving every fifth iteration thereafter. We assessed convergence via visual inspection of trace plots and defined adequate convergence as Gelman–Rubin convergence statistics ($\hat{R}$) < 1.1 [52, 55]. Regression coefficients whose 95% credible intervals (CrI) overlapped 0 were interpreted as uninformative. No permits or institutional research review were necessary to collect this data as it was only observational.

## Results

There was wide and representative coverage for all covariates, with the range of those measured by percentage (e.g., grass, woody, and VOR2) between 0–100 across all points (Table 1). Basal area ranged from 0–65 m$^2$/ha, though most points did not exceed ~40 m$^2$/ha. We did not include proportion forbs (%), proportion bare ground (%), and proportion litter (%) groundcover in modeling given their high degree of correlation (i.e., $|r| > 0.7$) with grass and woody covariates (S1 Table). We did not include horizontal visual obstruction at 0.25 m (%) or horizontal visual obstruction at 1 m above ground level (%), given their high degree of correlation (i.e., $|r| > 0.7$) with each other and VOR2. We did not include canopy closure (%) in the model, given the high degree of correlation (i.e., $|r| > 0.7$) with basal area. We justified this model specification given that these covariates would have confounded interpretation of slope coefficients and possibly resulted in overparameterization. Furthermore, parameters that were included in the model all had negligible correlation ($|r_{max}| < 0.35$) among each other (S1 Table), and in the case of basal area were common measurements in forest management. Comparing vegetation in treatment and control points (t-test, $P \leq 0.05$), treatment points had increased grass groundcover (grass [%]; $t_{1800} = 15.95$, $P \leq 0.001$), less woody groundcover (woody [%]; $t_{1800} = 9.16$, $P \leq 0.001$), less horizontal visual obstruction at 2 m above ground level (VOR2 [%]; $t_{1800} = 7.85$, $P \leq 0.001$), and decreased basal area (basal [m$^2$/ha]; $t_{1800} = 45.92$, $P \leq 0.001$) (Table 2).

We included 7,531 detections for the 12 priority focal species. After detections were binned into the two distance classes, detection functions did not appear to be appropriate, as clustering occurred at the farther of the two classes (i.e., 50–100 m); therefore, pooling across distance classes was performed. Markov chains reached convergence ($\hat{R}_{max} < 1.1$), and visual inspection

**Table 1. Mean, standard deviation, and range of all environmental covariates associated with hierarchical community abundance (N-mixture) model for select Bird Conservation Region (BCR) 30 priority breeding birds within the New Jersey Pine Barrens study area during 2012, 2013, 2016, and 2017, New Jersey, USA.** Treatment points received variable timber harvest and prescribed burning regimes prior to and during the study period, whereas reference (control) points did not receive such treatments and were characteristically representative of the predominantly forested landscape.

| Covariate [a] | Point type | $\bar{x}$ | SD | Range |
|---|---|---|---|---|
| Grass | Control | 0.28 | 1.64 | 0–15 |
| | Treatment | 12.69 | 24.49 | 0–100 |
| Woody | Control | 48.76 | 40.48 | 0–100 |
| | Treatment | 32.55 | 34.33 | 0–100 |
| VOR2 | Control | 47.66 | 27.40 | 0–100 |
| | Treatment | 36.56 | 32.67 | 0–100 |
| Basal | Control | 28.89 | 9.83 | 3.44–65.04 |
| | Treatment | 9.45 | 8.04 | 0–34.05 |

[a] Grass = proportion grass groundcover (%), Woody = proportion woody groundcover (%), VOR2 = horizontal visual obstruction at 2 m above ground level (%), Basal = basal area (m$^2$/ha).

**Table 2. Mean coefficients and derived parameters with 95% credible intervals (CrIs) from posterior distributions of hierarchical community abundance ($N$-mixture) model for select Bird Conservation Region (BCR) 30 priority breeding birds within the New Jersey Pine Barrens study area during 2012, 2013, 2016, and 2017.** Community- and BCR 30 suite-level parameters reported, as well as detection probability ($P_d$) and zero-inflation parameter ($\phi$). Bold denotes covariates with 95% CrIs that do not contain zero (i.e., a measure of significance). See text for covariate, parameter, and suite descriptions.

| | | Community-level parameters | Suite-level parameters | | |
| --- | --- | --- | --- | --- | --- |
| | | | Forested Upland | Scrub-shrub | Grassland |
| Detection | | | | | |
| | Intercept | **−1.71 (−2.27 to −1.23)** | **−1.70 (−2.50 to −1.03)** | **−1.61 (−2.34 to −1.00)** | **−2.27 (−2.99 to −1.71)** |
| | Date | **−0.40 (−0.79 to −0.03)** | **−0.48 (−0.79 to −0.21)** | **−0.44 (−0.67 to −0.22)** | **0.27 (0.15 to 0.38)** |
| | Date$^2$ | **−0.18 (−0.37 to 0.00)** | −0.22 (−0.47 to 0.01) | **−0.20 (−0.36 to −0.05)** | **0.18 (0.08 to 0.27)** |
| | Wind | **−0.14 (−0.29 to −0.04)** | −0.15 (−0.42 to 0.03) | **−0.14 (−0.36 to 0.00)** | −0.02 (−0.16 to 0.19) |
| | Disturbance | 0.00 (−0.11 to 0.09) | −0.01 (−0.20 to 0.14) | 0.01 (−0.13 to 0.14) | −0.02 (−0.20 to 0.10) |
| Abundance [a] | | | | | |
| | Grass | 0.04 (−0.03 to 0.10) | 0.05 (−0.08 to 0.17) | 0.02 (−0.08 to 0.12) | 0.06 (−0.02 to 0.13) |
| | Woody | −0.17 (−0.42 to 0.07) | −0.14 (−0.42 to 0.13) | **−0.21 (−0.43 to −0.01)** | −0.11 (−0.28 to 0.07) |
| | VOR2 | −0.08 (−0.25 to 0.08) | −0.10 (−0.34 to 0.12) | −0.04 (−0.21 to 0.11) | **−0.12 (−0.24 to 0.00)** |
| | Basal | **−0.23 (−0.40 to −0.09)** | −0.19 (−0.46 to 0.07) | **−0.26 (−0.51 to −0.06)** | **−0.37 (−0.66 to −0.13)** |
| | Treatment | **0.44 (0.20 to 0.76)** | 0.07 (−0.11 to 0.45) | **0.59 (0.48 to 0.75)** | **1.83 (1.71 to 2.09)** |
| | Site (random) | 0.00 (−0.23 to 0.24) | – | – | – |
| Derived parameters | | | | | |
| | $P_d$ | 0.16 (0.14 to 0.19) | 0.16 (0.09 to 0.24) | 0.17 (0.11 to 0.26) | 0.12 (0.06 to 0.18) |
| | $\phi$ | 0.56 (0.29 to 0.86) | 0.31 (0.08 to 0.67) | 0.61 (0.42 to 0.83) | 0.25 (0.10 to 0.44) |

[a] Grass = proportion grass groundcover (%), Woody = proportion woody groundcover (%), VOR2 = horizontal visual obstruction at 2 m above ground level (%), Basal = basal area (m$^2$/ha).

of trace plots indicated mixing among chains. Community-level mean detection during the 10-min surveys ($P_d$) was 0.16 (CrI = [0.14, 0.19]) and was negatively influenced by date (quadratic, concave down) and wind speed (Table 2). Combining species into their respective suite-level habitats, Scrub-Shrub species ($n$ = 5) showed a similar response to detection, whereas Forested Upland species ($n$ = 6) only showed a linear (negative) effect of date on detection. Conversely, Grassland species ($n$ = 1) mean detectability was positively influenced by date. Zero-inflation parameter estimates ($\phi$) ranged from 0.25 (CrI = [0.10, 0.44]) for Grassland species to 0.61 (CrI = [0.42, 0.83]) for Scrub-Shrub species.

At the community level, abundance was greater at treatment points compared to control points (Table 2). Additionally, we observed a negative association with basal area (m$^2$/ha); predicted community size increased by 1 bird (CrI = 0.39–1.74) for every 26 m$^2$ decrease in basal area per 2.17 ha. Community size did not exhibit a strong relationship with any additional vegetation covariates (Table 2).

At the BCR 30 habitat suite level, Forested Upland species as a group did not exhibit a significant directional relationship with any environmental parameter; however, individual species within this suite did show such relationships. Baltimore oriole, northern flicker, and wood thrush abundance were negatively associated with proportion woody groundcover, while black-and-white warbler abundance showed a positive association with woody groundcover (Fig 1). Black-and-white warbler abundance also showed a positive relationship with horizontal visual obstruction at 2 m above ground level, whereas great crested flycatcher abundance exhibited a negative trend. Five of the six Forested Upland species showed a weak negative association with basal area, but only norther flickers confidence interval did not overlap zero.

Scrub-Shrub species as a group showed a negative association with proportion woody groundcover and basal area, as well as an overall greater abundance at treatment points

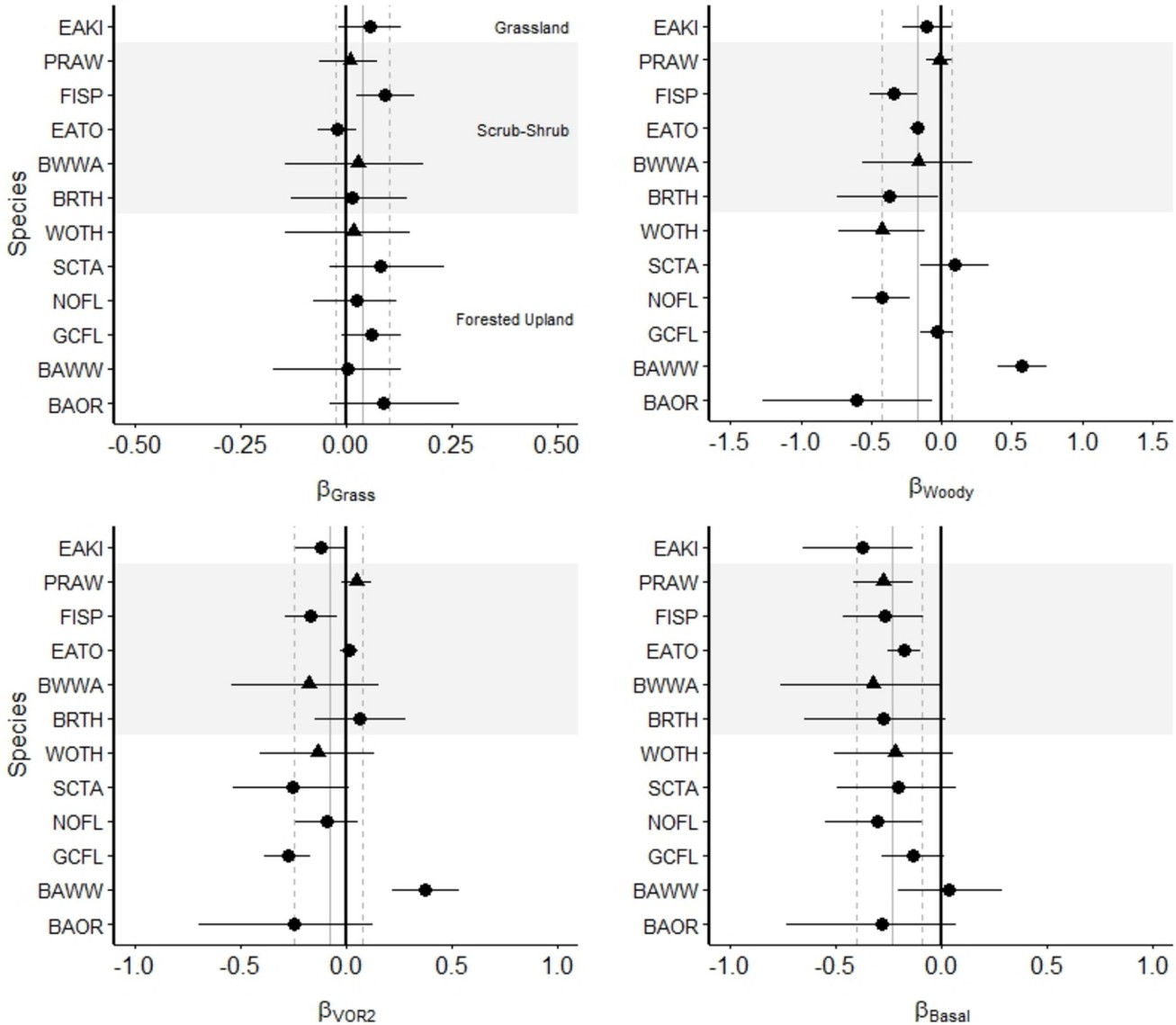

**Fig 1. Comparison of breeding bird community, Bird Conservation Region (BCR) 30 species-suite (Forested Upland, Scrub-Shrub, and Grassland), and individual species response in abundance at 100-m-radius point counts for four scaled environmental covariates, representing three strata levels: Groundcover (proportion grass [$\beta_{Grass}$] and proportion woody [$\beta_{Woody}$]), midstory (horizontal visual obstruction at 2 m above ground level [$\beta_{VOR2}$]), and canopy (basal area m²/ha [$\beta_{Basal}$]) within the New Jersey Pine Barrens during 2012, 2013, 2016, and 2017, New Jersey, USA.** Forested Upland species include Baltimore oriole (BAOR), black-and-white warbler (BAWW), great crested flycatcher (GCFL), northern flicker (NOFL), scarlet tanager (SCTA), and wood thrush (WOTH); Scrub-Shrub species include blue-winged warbler (BWWA), brown thrasher (BRTH), eastern towhee (EATO), field sparrow (FISP), and prairie warbler (PRAW); and one Grassland species, eastern kingbird (EAKI). Points show species-level posterior mean and 95% credible interval (CrI), with BCR 30 priority designation indicated by triangles (highest) and circles (high). Gray vertical lines show posterior mean (solid) and 95% CrI (dashed) of the community mean hyperparameter. When 95% CrI did not overlap 0 (black vertical line), it was interpreted as a significant effect. Note differences in scale among slope coefficient estimates. See text for species codes.

compared to control points. Two notable relationships within this suite were a positive association between field sparrow abundance and proportion grass groundcover, and the strong negative association between prairie warbler (designated "highest" priority species) abundance and basal area.

Grassland species were represented only by the eastern kingbird. Eastern kingbird abundance was negatively associated with horizontal visual obstruction at 2 m above ground level and basal area, with the latter being among the strongest effects reported (Table 2). Additionally, eastern kingbird abundance increased by a magnitude ~3 times greater than that observed by Scrub-Shrub species at treatment points compared to control points.

Species-level Jaccard co-occurrence indices ranged from 0.03 (CrI = [0.00, 0.13]) for Baltimore oriole and blue-winged warbler to 0.76 (CrI = [0.66, 0.83]) for eastern towhee and prairie warbler (Fig 2). Eastern towhees were the most ubiquitous species, exhibiting a high degree of co-occurrence (i.e., $J_{ij}$ CrI > 0.5) with three Forested Upland species and two Scrub-Shrub species. For the BCR 30 highest priority species, blue-winged warbler and wood thrush, degree of co-occurrence was low (i.e., all $J_{ij}$ < 0.2) with all other species; however, for prairie warblers, a high degree of co-occurrence was observed with species representative of all three species-suites: Forested Upland (great crested flycatcher, northern flicker, and black-and-white warbler), Scrub-Shrub (eastern towhee and field sparrow), and Grassland (eastern kingbird). Eastern kingbird showed a moderate degree of co-occurrence with four of the five Scrub-Shrub species, and a low degree of co-occurrence with four of the six Forested Upland species.

## Discussion

Atlantic Coastal Plain pine-dominated ecosystems evolved with fire-tolerance and fire-dependence [14, 15]; however, fire suppression has decreased fire-mediated land area [15, 16] as well as species composition and structure [17–22]. Our goal was to explore whether the impacts of forest management through thinning and burning could support more occurrence and abundance of high priority bird species [7–11] in the New Jersey Pine Barrens. We found evidence that reducing live tree basal area and promoting diverse early successional groundcover within the NJPB (thus creating pine savannahs) can collectively benefit multiple suites of regionally significant upland breeding birds. Species that occupied dense (e.g., > 20 $m^2$/ha) pine stands, representative of much of the NJPB, existed at comparable abundances in reduced basal area stands, while grassland and early successional-obligate species increased in abundance in the pine savannahs. These results are well documented in the literature where reductions in basal density through burning and thinning can increase bird abundance and biodiversity [9 for review].

The Forested Upland species suite of priority species for BCR30 was the largest and most diverse among those studied, including species traditionally associated with pine-dominant, hardwood-dominant, and mixed upland forests [29]. The underrepresentation of BCR 30 hardwood-associated species was anticipated, given the focus on upland pine management. For example, infrequent observation of Baltimore oriole and wood thrush was likely a result of common association with riparian wooded edges [56] and closed canopy hardwood forests [57], respectively. Accordingly, conclusions based on upland pine management are not necessarily applicable to these species and we cannot assume that stand thinning and prescribed burning will not negatively impact their abundance.

Given the diverse habitat requirements among Forested Upland species, some management practices may result in tradeoffs. For example, Baltimore oriole, northern flicker, wood thrush, and black-and-white warbler showed a mixed response to woody groundcover likely due to natural differences in niche portioning within habitat preferences. Additionally, black-and-white warbler and great crested flycatcher showed strong but opposite associations with horizontal visual obstruction at 2 m above ground level. Such tradeoffs between BCR 30 priority species, even within a single habitat suite, favors fine-scale over coarse-scale management application within the NJPB and should assist decision makers with quantitative species-level

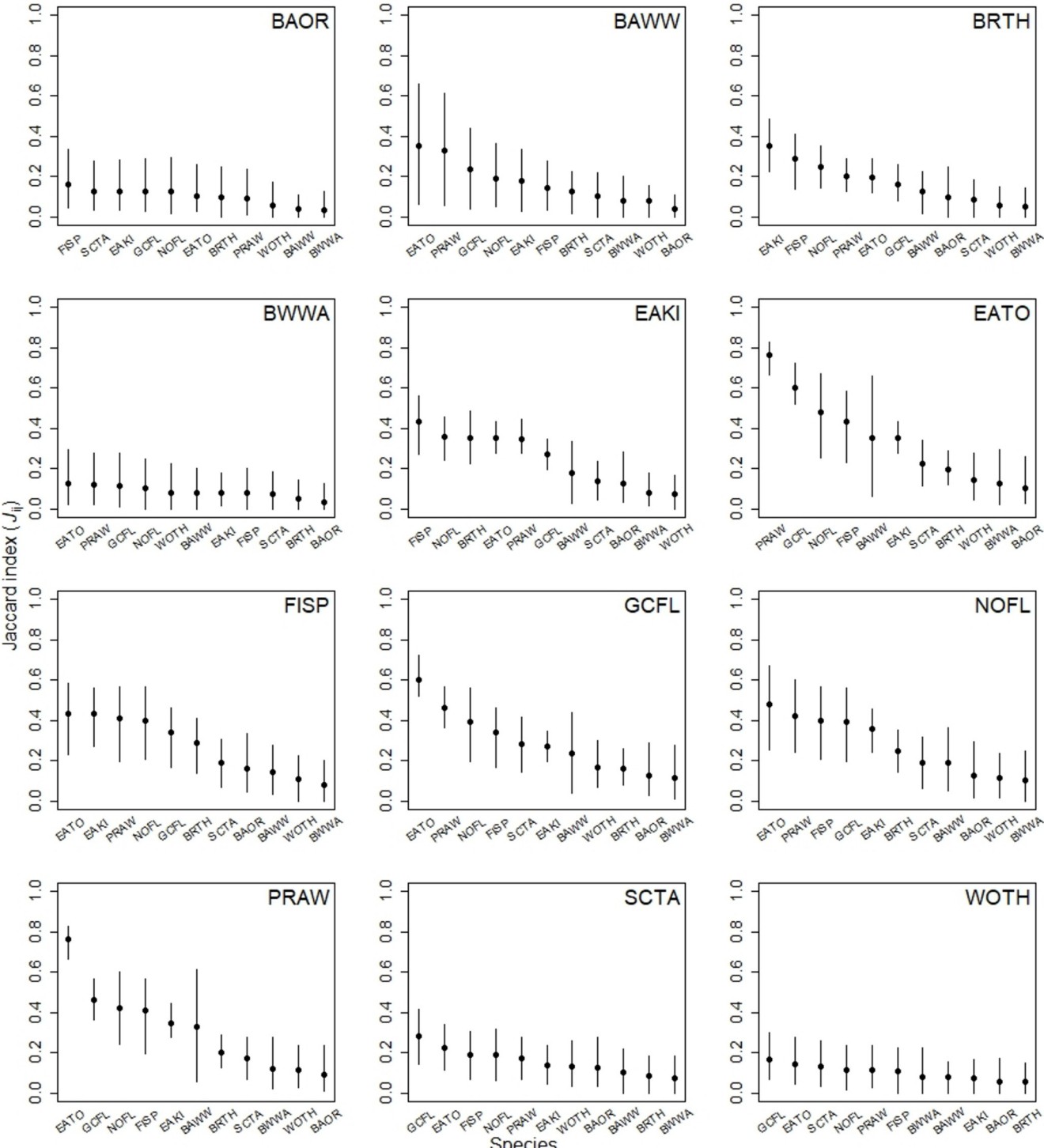

**Fig 2. Jaccard indices ($J_{ij}$) for all pair-wise comparisons among Bird Conservation Region (BCR) 30 priority species ($n$ = 12) from 100-m-radius point counts within the New Jersey Pine Barrens during 2012, 2013, 2016, and 2017., New Jersey, USA.** Values of $J_{ij}$ range from 0 to 1, corresponding respectively to the extremes of no co-occurrence and perfect co-occurrence. Each plot represents respective co-occurrence probabilities for the species of interest (top-right corner) among all count locations ($n$ = 150) against the remaining species in descending order of co-occurrence. See text for species codes.

alternative management actions. Further, recent work also suggests that some forest interior nesting species shift their habitat to early-successional/young forests post-breeding due to the availability of forage and cover [2, 6, 58, 59] and the need to fulfill energy requirements of development or molt [13, 60–62]. Therefore, varying responses to forest management was not unexpected, and further supports the need for heterogeneity and diversity in forest structure and vegetative composition.

A reduction in basal area from the densities found throughout much of the NJPB increased mean abundance of BCR 30 Scrub-Shrub/Early Successional/Young Forest priority birds. This species suite is traditionally associated with open forests, shrub-scrub lands, grasslands, and agricultural landscapes where diverse early successional groundcover predominates [63] thus producing more open nesting opportunities and more insect and seed food resources preferred by these species [9]. Management of groundcover via frequent (<3-year fire return interval) prescribed burning or mechanical treatment typically produces ground-level vegetation communities that provide forage and cover for Scrub-Shrub associated species [64]. Dense basal areas and associated closed canopies may preclude establishment of groundcover, given the relatively high amount of sunlight required by heliophytic plants (e.g., grasses and forbs). A majority of the current NJPB landscape is densely stocked pitch and shortleaf pines [25] characteristically unconducive to Scrub-Shrub species. As such, reduction in basal area and complementary management of groundcover (e.g., burning, herbicide, or mechanical treatment) should improve occupancy and increase abundance of these species experiencing range-wide declines.

The two highest priority Scrub-Shrub species, blue-winged warbler and prairie warbler, both showed a positive association with basal area reduction. Between 1966–2015, blue-winged warbler and prairie warbler have declined 2.9%/year and 3.1%/year, respectively, within BCR 30 [65]. These regional declines reflected the concurrently observed afforestation and forest succession within the NJPB [25]; however, the estimated magnitude of these species declines were not reflected at the point count level. Specifically, we observed a high occurrence of prairie warblers at both treatment and control sites, as well as a high co-occurrence (Jaccard Indices) between prairie warblers and species representative of all three BCR30 species suites; neither of these patterns were observed for blue-winged warblers. The comparatively high prevalence of prairie warblers at both treatment and control sites may be indicative of their resiliency to persist within non-optimal forest conditions (closed canopy, but have high woody shrub components and low percent grass cover) across the massive NJPB landscape. Morimoto and Wasserman [66] found prairie warblers were less abundant in areas with increased percent cover and basal area of pitch pines, and areas characterized by high oak tree cover. Previous research in the NJPB indicated prairie warbler occupancy was higher in areas with increased vegetation at 2m and decreased grass and canopy cover [67]. Another typical habitat characteristic of prairie warblers is open Scrub-Shrub areas that contain poor quality, dry, and sandy soil [68], which is the most common soil type in the NJPB. This coincides with the positive association of prairie warblers and understory vegetation because as the amount of shrubby vegetation increases, sunlight reaching the forest floor decreases, as does the growth of grass [67]. Prairie warblers within pine barrens in Massachusetts have been shown to require larger patches (e.g., clear cuts) than other early-successional birds to confer site-level occupancy [69, 70]. Thus, within the NJPB, prairie warbler abundance reflects the suitability of site features (e.g., basal area and vegetation species composition), as well as overall landscape area.

Grassland species inherently require low (or zero) basal areas and early successional groundcover to meet their ecological needs for nesting and food. The only BCR 30 Grassland species represented here was the eastern kingbird, so conclusions regarding management for the entire suite should be limited. Eastern kingbirds are typically associated with grasslands

and shrublands [71], though they show a moderate tolerance to habitat alteration [72]. In forested areas, eastern kingbirds show a proclivity to nest in frequently burned patches [73], which tend to be more open. The high degree of co-occurrence between eastern kingbirds and field sparrows ($J_{EAKI,FISP} > 0.4$) is an indication that these species have similar habitat requirements and management for one will likely benefit the other; however, scale of management is an important consideration. Multi-scale analyses have shown that eastern kingbirds are closely associated with landscape-level composition, while field sparrows were more closely related to patch-scale factors [74]. Therefore, while site-level associations on abundance have been quantified here, additional landscape consideration may be needed. Furthermore, research should explicitly test the influence of grassland creation on extirpated grassland-associated species in the NJPB.

Through active forest management that promotes diversity in forest age class and native vegetative species composition, many other wildlife and natural resource objectives will be met. Ecosystems with high structural heterogeneity and biodiversity are more resilient to human stressors by absorbing carbon dioxide and nitrogen better than forests with lower biodiversity [75]. Diverse forest ecosystems containing native plants and mixed age classes face lower risk of failure than simplified systems, because such complex communities have built-in redundancies [76]. These resilient ecosystems offer mixtures of uneven-aged tree stands that provide a vertical diversity of vegetation structure that can enhance habitat conditions for many of wildlife species. Managing forest conditions in both younger and older age classes (and smaller and larger structural stages) to maintain both early and late successional habitats for a diversity of forest associated species may also conserve habitat and viable populations of many forest-associated wildlife species [77].

We recommend an increase in active forest stewardship for upland birds that: 1) aims to increase heterogeneity in basal area and canopy tree age class; 2) promotes increased biodiversity through a mixed composition of woody, forb, and warm season grass groundcover (e.g., increased diversity of native vegetation species composition); and 3) is implemented at-scale to reduce wildfire fuel loads within the NJPB. Specifically, we recommend targeting basal areas between ~0–15 m$^2$/ha via selective thinning, shelter cutting, and small-scale clear cutting of the pine forests to meet these objectives. The most economically feasible way to create and maintain early seral stage groundcover would be a combination of mechanical treatment (e.g., mowing, roller chopping, etc.) and prescribed burning, which would have the added benefit of reducing fuel loads across this ~4,500 km$^2$ landscape [26]. Additionally, by planning for and implementing active forest stewardship, these actions are also in line with the 2020 NJ Forest Action Plan that calls for addressing climate resiliency of our forest through implementing carbon defense strategies in the NJPB.

## Supporting information

**S1 Table. Pairwise correlations between all environmental variables to characterize songbird presence including % grass, % forbes, % woody, % litter, % bare ground, basal density, canopy coverage, and visual obstruction at 0.25, 1, and 2 m.** Bolded values indicate those value above our cutoff of $|r| = 0.7$.
(DOCX)

## Acknowledgments

We would like to thank P. Coppola for data collection and analysis.

## Author Contributions

**Conceptualization:** Christopher K. Williams, John Parke.

**Data curation:** Christopher K. Williams, Elizabeth A. Matseur.

**Formal analysis:** Theron M. Terhune, II.

**Funding acquisition:** Christopher K. Williams, John Parke, John Cecil.

**Investigation:** Christopher K. Williams, John Parke.

**Methodology:** Christopher K. Williams, Theron M. Terhune, II, John Parke.

**Project administration:** Christopher K. Williams, John Parke, John Cecil.

**Resources:** Christopher K. Williams.

**Supervision:** Christopher K. Williams.

**Validation:** Christopher K. Williams.

**Writing – original draft:** Christopher K. Williams, Theron M. Terhune, II, John Parke, John Cecil.

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
