## [Decision Letter · Decision Letter 0]

17 Jul 2023

PONE-D-23-17857Active forest stewardship benefits priority birds in the New Jersey Pine BarrensPLOS ONE

Dear Dr. Williams,

Thank you for submitting your manuscript to PLOS ONE. After careful consideration, we feel that it has merit but does not fully meet PLOS ONE’s publication criteria as it currently stands. Therefore, we invite you to submit a revised version of the manuscript that addresses the points raised during the review process.

I have received comments from two reviewers. Based on their assessments as well as my own reading, I believe your manuscript could be suitable for publication with a moderate level of revision and restructuring. In particular, I agree with Reviewer #1 that your manuscript would benefit from better linkage of your study to theory. Your manuscript is well written, to be sure, but it doesn't tie the study system (i.e., the New Jersey Pine Barrens) to other examples of forest stewardship in historically fire-dependent systems and the response of breeding bird communities. Reviewer #1 has provided a short list of references but there are certainly others available from refereed literature. I believe your manuscript will find a more receptive audience and thus have a greater impact if you couch it within a broader ecological context. I don't think that this will necessitate a huge number of changes, just some additional text in the Introduction and Discussion sections as well as some shifting of material between the Introduction and the first part of the Methods section (more or less along the lines proposed by Reviewer #1.) Additionally, you should consider the analytical suggestions of Reviewer #1, especially those related to possible covariates from the vegetation data.

We look forward to receiving your revised manuscript.

Kind regards,

Frank H. Koch, PhD

Academic Editor

PLOS ONE

Journal Requirements:

Additional Editor Comments (if provided):

Line 127 - delete colon before "scrub oaks"

Line 133 - delete comma after "including"

Line 348 - "assist" instead of "assists"

“CKW. 05012015. New Jersey Audubon. https://njaudubon.org/. The sponsors assisted with study design and preparation of the manuscript.”

“We would like to thank P. Coppola for data collection and New Jersey Audubon, Pine Creek Forestry, New Jersey Division of Fish and Wildlife, Haines Family Foundation, Peter R. & Cynthia K. Kellogg Foundation, and many other private donors for funding this research.  Additionally this research was funded by USDA Hatch (DEL00774) and the University of Delaware Waterfowl and Upland Gamebird Center.”

“CKW. 05012015. New Jersey Audubon. https://njaudubon.org/. The sponsors assisted with study design and preparation of the manuscript.”

6. We note that you have indicated that data from this study are available upon request. PLOS only allows data to be available upon request if there are legal or ethical restrictions on sharing data publicly. For more information on unacceptable data access restrictions, please see http://journals.plos.org/plosone/s/data-availability#loc-unacceptable-data-access-restrictions.

8. We note that Figure 1 in your submission contain [map/satellite] images which may be copyrighted. All PLOS content is published under the Creative Commons Attribution License (CC BY 4.0), which means that the manuscript, images, and Supporting Information files will be freely available online, and any third party is permitted to access, download, copy, distribute, and use these materials in any way, even commercially, with proper attribution. For these reasons, we cannot publish previously copyrighted maps or satellite images created using proprietary data, such as Google software (Google Maps, Street View, and Earth). For more information, see our copyright guidelines: http://journals.plos.org/plosone/s/licenses-and-copyright.

Reviewers' comments:

Reviewer's Responses to Questions

**Comments to the Author**

1. Is the manuscript technically sound, and do the data support the conclusions?

Reviewer #1: Partly

Reviewer #2: Yes

2. Has the statistical analysis been performed appropriately and rigorously? 

Reviewer #1: Yes

Reviewer #2: Yes

3. Have the authors made all data underlying the findings in their manuscript fully available?

Reviewer #1: Yes

Reviewer #2: Yes

4. Is the manuscript presented in an intelligible fashion and written in standard English?

Reviewer #1: Yes

Reviewer #2: Yes

5. Review Comments to the Author

Reviewer #1: Dear Editor,

I found the manuscript ‘Active forest stewardship benefits priority birds in the New Jersey Pine Barrens’ an interesting one and I believe it could be published after the authors have addressed some major comments.

The introduction does not provide much information about the theory behind the research. The effects of forest management on birds, the role of fire management and the different responses of bird species are not mentioned. In contrast, lot of text is devoted to describing the study area and the study system, which in my opinion should be moved to the method section (e.g., L59-102).

Similarly, the discussion is focused only on the interpretation of the results but misses to put them into a perspective that could favor comparison with other systems and translation. I strongly suggest the authors to revise thoroughly, discuss where their results stand in relation to what is known about bird responses and include comparison with more international references. Below, I provide a list of publication that could be used although more is available in the literature:

Response to forest management

https://doi.org/10.3 832/ifor1212-007

https://doi.org/10.1186/s12915-021-01136-8 (particularly for basal area)

Response to fire management

Kotliar et al. Studies in Avian Biology No. 25:49-64, 2002.

https://doi.org/10.1007/s40725-022-00175-w

Conservation and forest management

https://doi.org/10.1016/j.foreco.2016.10.008

https://doi.org/10.1111/j.1523-1739.2007.00756.x

L76: What about this motivation? Can you say more?

L79: Why is mentioning mammals relevant? I suggest removing.

L81-89: Many organizations are mentioned here but I believe they mean nothing to an international audience. I also believe this text is not relevant to an introduction and should be reduced and moved to the methods.

L100: Do you mean ‘principles’?

L102: Not clear what is the suite-level at this stage of the manuscript.

L139: I guess that 100 m is the length of the fixed-radius, right?

L149: I suggest to use methodological references here in addition or in place of the current ones, as Dawson and Efford is a technical paper and Ladin et al is not a methodological paper. I suggest to mention Bibby’s Bird Census Techniques as a general reference, and more specific to forests could be https://doi.org/10.3161/00016454AO2017.52.1.001

L154: I suggest to test the time of the sampling as a further detection covariate. Birds, especially songbirds, don’t start singing at the same time in the morning, but in a rather predictable way depending on species. Some species don’t start before sunrise whereas other starts well before e.g., https://doi.org/10.1002/ece3.9491 In alternative, the authors should explain how they minimized this potential source of bias.

Figure 1: I suggest to use letters for each figure panel and to refer to each panel in the caption. The second panel is not very informative: it is not clear what are the treatments and control and how distributed are the sites and the points. In the methods it is said that points are space >300 m but the caption says that minimum distance was 200 m, which one is correct?

L182: Not clear what is meant here by ‘less intrusive and not targeted’.

Vegetation Measurements: It should be clarified that the vegetation data was collected every year. In addition, did you separate live from dead trees? What about lying deadwood? These are variables known to influence birds. Could you include them in your analysis?

L203: Why did you include a site random effect and not also a year random effect? Would it be wise to include that in the analysis?

L230: Here it is stated that you restricted your analysis to territorial breeding birds but in the objectives (L103-104) it is stated that the targets are 12 species. This may be confusing for the readers. Please, clarify.

L277-280: I assume that the figures here reported are from a t-test of vegetation variables between treatments and controls. however, this has not been described in the methods nor it is clear from these lines. Please, include a full description of all analysis.

L281: It is not clear how many species were included in the model. Form here, I would say only 12 species. It is also not clear how were suite-level estimates derived. Were the estimates for each species within the same suite pooled together or else? Does this mean that the estimated for grassland are actually the estimates for one species?

Reviewer #2: This is a very interesting article, one that is well-written and presented. The scale of the effort to collect and interpret these data is also impressive, providing what appear to be a sound assessment of species interactions with vegetation structure within the geographic area of focus. I do not perceive any major issues affecting the publication of this article. However, I did not several relatively trivial items that can be addressed before publication.

In the abstract (line 37) you note the suite of species evaluated as 'N' but refer elsewhere to the suite of species as 'n' (line 253). Not clear why the two different symbols are used.

Line 163 "Jun-Jul" should be separated with a long dash as have been other spatial spans throughout paper.

Line 290: you reference the effect for "Grassland Species" but my impression is that this suite is reflected by a single species, the Eastern Kingbird. Perhaps to clarify this you should reference this suite of species in the results as represented by this one species. This could be done similarly to what you have on line 310.

Line 316: Why is the alpha code for Blue-winged Warbler presented here but the appropriate has not been presented for other species at first mention? Looking at Figs. 2 and species on x- and y-axis (Fig.2 and Fig. 3) are presented as the alpha codes. It is not clear however where these alpha codes are defined, outside of this case for the Blue-winged Warbler, outside text lines 93+. It may be most efficient to include a table of the species, alpha codes, and reflective PIF indices for NJ, and/or the appropriate geographic conservation area.

Lines 352-355: I think it could also be argued for inclusion of more refined temporal evaluations in your analysis and when the surveys were conducted. For instance, if these birds go to denser areas during the post-fledging period it would be interesting to assess species-structure response at various times within the sampling framework to see if there is a shift in the structure of the conditions used.

6. PLOS authors have the option to publish the peer review history of their article (what does this mean?). If published, this will include your full peer review and any attached files.

Reviewer #1: No

Reviewer #2: No

---

## [Author Response · Author response to Decision Letter 0]

21 Dec 2023

Recommendations from Associate Editor

In particular, I agree with Reviewer #1 that your manuscript would benefit from better linkage of your study to theory. Your manuscript is well written, to be sure, but it doesn't tie the study system (i.e., the New Jersey Pine Barrens) to other examples of forest stewardship in historically fire-dependent systems and the response of breeding bird communities. Reviewer #1 has provided a short list of references but there are certainly others available from refereed literature. I believe your manuscript will find a more receptive audience and thus have a greater impact if you couch it within a broader ecological context. I don't think that this will necessitate a huge number of changes, just some additional text in the Introduction and Discussion sections as well as some shifting of material between the Introduction and the first part of the Methods section (more or less along the lines proposed by Reviewer #1.) Additionally, you should consider the analytical suggestions of Reviewer #1, especially those related to possible covariates from the vegetation data.

Response: Fixed

Line 127 - delete colon before "scrub oaks"

Response: Done

Line 133 - delete comma after "including"

Response: Done

Line 348 - "assist" instead of "assists"

Response: Done

In your Methods section, please provide additional information regarding the permits you obtained for the work. Please ensure you have included the full name of the authority that approved the field site access and, if no permits were required, a brief statement explaining why.

Response: Done

Thank you for stating the following financial disclosure:

“CKW. 05012015. New Jersey Audubon. https://njaudubon.org/. The sponsors assisted with study design and preparation of the manuscript.”

5. Thank you for stating the following in the Acknowledgments Section of your manuscript: “We would like to thank P. Coppola for data collection and New Jersey Audubon, Pine Creek Forestry, New Jersey Division of Fish and Wildlife, Haines Family Foundation, Peter R. & Cynthia K. Kellogg Foundation, and many other private donors for funding this research. Additionally this research was funded by USDA Hatch (DEL00774) and the University of Delaware Waterfowl and Upland Gamebird Center.”

We note that you have provided funding information that is currently declared in your Funding Statement. However, funding information should not appear in the Acknowledgments section or other areas of your manuscript. We will only publish funding information present in the Funding Statement section of the online submission form. Please remove any funding-related text from the manuscript and let us know how you would like to update your Funding Statement. Currently, your Funding Statement reads as follows: “CKW. 05012015. New Jersey Audubon. https://njaudubon.org/. The sponsors assisted with study design and preparation of the manuscript.”

Response: Done

6. We note that you have indicated that data from this study are available upon request. PLOS only allows data to be available upon request if there are legal or ethical restrictions on sharing data publicly. For more information on unacceptable data access restrictions, please see http://journals.plos.org/plosone/s/data-availability#loc-unacceptable-data-access-restrictions.

Response: The data is now publicly available at: https://drive.google.com/drive/folders/1yf9HmjW5aQFrAK7bLowyeRoY3AsnLgAC?usp=sharing

Response: Done

8. We note that Figure 1 in your submission contain [map/satellite] images which may be copyrighted. All PLOS content is published under the Creative Commons Attribution License (CC BY 4.0), which means that the manuscript, images, and Supporting Information files will be freely available online, and any third party is permitted to access, download, copy, distribute, and use these materials in any way, even commercially, with proper attribution. For these reasons, we cannot publish previously copyrighted maps or satellite images created using proprietary data, such as Google software (Google Maps, Street View, and Earth). For more information, see our copyright guidelines: http://journals.plos.org/plosone/s/licenses-and-copyright.

Response: None of the images in Figure 1 are copywrited and all were created by the authors inside ArcView GIS software therefore we have left Figure 1 as is.

Reviewer #1: 

The introduction does not provide much information about the theory behind the research. The effects of forest management on birds, the role of fire management and the different responses of bird species are not mentioned. In contrast, lot of text is devoted to describing the study area and the study system, which in my opinion should be moved to the method section (e.g., L59-102). Similarly, the discussion is focused only on the interpretation of the results but misses to put them into a perspective that could favor comparison with other systems and translation. I strongly suggest the authors to revise thoroughly, discuss where their results stand in relation to what is known about bird responses and include comparison with more international references. Below, I provide a list of publication that could be used although more is available in the literature:

Response to forest management

https://doi.org/10.3 832/ifor1212-007

https://doi.org/10.1186/s12915-021-01136-8 (particularly for basal area)

Response to fire management

Kotliar et al. Studies in Avian Biology No. 25:49-64, 2002.

https://doi.org/10.1007/s40725-022-00175-w

Conservation and forest management

https://doi.org/10.1016/j.foreco.2016.10.008

https://doi.org/10.1111/j.1523-1739.2007.00756.x

Response: Thank you for the recommendation and we agree. We have added additional language to the introduction and discussion to provide more context. Additionally, after thinking about different ways to reduce the background story of the New Jersey Pine Barrens in the Introduction and move it to the study area description, we ultimately decided it was more important to keep in in the introduction as it provides a valuable historical context to our story.

L76: What about this motivation? Can you say more?

Response: We rewrote this sentence

L79: Why is mentioning mammals relevant? I suggest removing.

Response: We believe it was important to state this to give context to previous research in the New Jersey Pine Barrens and to highlight a major weakness in another taxa group: birds. Therefore, we have decided to leave this comparison in the introduction.

L81-89: Many organizations are mentioned here but I believe they mean nothing to an international audience. I also believe this text is not relevant to an introduction and should be reduced and moved to the methods.

Response: Fixed

L100: Do you mean ‘principles’?

Response: Fixed

L102: Not clear what is the suite-level at this stage of the manuscript.

Response: Clarified in the preceding sentences.

L139: I guess that 100 m is the length of the fixed-radius, right?

Response: Correct, and we believe the grammar of the sentence is correct to express that.

L149: I suggest to use methodological references here in addition or in place of the current ones, as Dawson and Efford is a technical paper and Ladin et al is not a methodological paper. I suggest to mention Bibby’s Bird Census Techniques as a general reference, and more specific to forests could be https://doi.org/10.3161/00016454AO2017.52.1.001

Response: We added the Bibby reference but skipped the Balestrieri reference since the methodology differed slightly. 

L154: I suggest to test the time of the sampling as a further detection covariate. Birds, especially songbirds, don’t start singing at the same time in the morning, but in a rather predictable way depending on species. Some species don’t start before sunrise whereas other starts well before e.g., https://doi.org/10.1002/ece3.9491 In alternative, the authors should explain how they minimized this potential source of bias.

Response: Thank you for this recommendation. Our goal is to focus on presence/absence in the breeding season at a broad level. By conducting surveys in a randomized fashion, we hoped to observe birds across a variable sets of times thus capturing the imperfect detection variability probability inherently built into the detection probability of the equation. That being said, we have added this referenced in our methods to acknowledge this issue. 

Figure 1: I suggest to use letters for each figure panel and to refer to each panel in the caption. The second panel is not very informative: it is not clear what are the treatments and control and how distributed are the sites and the points. In the methods it is said that points are space >300 m but the caption says that minimum distance was 200 m, which one is correct?

Response: Figure updated with lettered panels and thank you for catching the 300m reference in the methods. That should have said 200m and we have corrected the methods.

L182: Not clear what is meant here by ‘less intrusive and not targeted’.

Vegetation Measurements: It should be clarified that the vegetation data was collected every year. In addition, did you separate live from dead trees? What about lying deadwood? These are variables known to influence birds. Could you include them in your analysis?

Response: We have rewritten the section to address your concerns. Unfortunately we can not discern the difference in life versus dead woody vegetation from the original data collection but noted both were included in wood vegetation.

L203: Why did you include a site random effect and not also a year random effect? Would it be wise to include that in the analysis?

Response: We appreciate this observation and had made the decision to remove time as a random variable as it was causing cofounding model performance and felt confident in this course of action as earlier explorative models of random effects of time showed it had very low significance. We have included additional language in our methods to explain this.

L230: Here it is stated that you restricted your analysis to territorial breeding birds but in the objectives (L103-104) it is stated that the targets are 12 species. This may be confusing for the readers. Please, clarify.

Respond: Fixed

L277-280: I assume that the figures here reported are from a t-test of vegetation variables between treatments and controls. however, this has not been described in the methods nor it is clear from these lines. Please, include a full description of all analysis.

Response: Added

L281: It is not clear how many species were included in the model. Form here, I would say only 12 species. It is also not clear how were suite-level estimates derived. Were the estimates for each species within the same suite pooled together or else? Does this mean that the estimated for grassland are actually the estimates for one species?

Response: We have attempted to clarify

Reviewer #2: 

In the abstract (line 37) you note the suite of species evaluated as 'N' but refer elsewhere to the suite of species as 'n' (line 253). Not clear why the two different symbols are used.

Response: We have corrected that to a lower case “n” to match all the other instances.

Line 163 "Jun-Jul" should be separated with a long dash as have been other spatial spans throughout paper.

Response: Fixed

Line 290: you reference the effect for "Grassland Species" but my impression is that this suite is reflected by a single species, the Eastern Kingbird. Perhaps to clarify this you should reference this suite of species in the results as represented by this one species. This could be done similarly to what you have on line 310.

Response: Fixed

Line 316: Why is the alpha code for Blue-winged Warbler presented here but the appropriate has not been presented for other species at first mention? Looking at Figs. 2 and species on x- and y-axis (Fig.2 and Fig. 3) are presented as the alpha codes. It is not clear however where these alpha codes are defined, outside of this case for the Blue-winged Warbler, outside text lines 93+. It may be most efficient to include a table of the species, alpha codes, and reflective PIF indices for NJ, and/or the appropriate geographic conservation area.

Response: I removed the blue winged warbler AOU code from the results. All of the other codes are listed up in the Introduction. But I am also sensitive to the fact that Figures should be self-descriptive. Therefore, I have added the descriptions to the codes in the Figure description.

Lines 352-355: I think it could also be argued for inclusion of more refined temporal evaluations in your analysis and when the surveys were conducted. For instance, if these birds go to denser areas during the post-fledging period it would be interesting to assess species-structure response at various times within the sampling framework to see if there is a shift in the structure of the conditions used.

Response: Thank you for that suggestion. While we are not in a position to reanalyze our data at smaller time scales, we made sure to add that recommendation in the discussion for future researchers to examine.

---

## [Decision Letter · Decision Letter 1]

12 Feb 2024

PONE-D-23-17857R1Active forest stewardship benefits priority birds in the New Jersey Pine Barrens

PLOS ONE

Dear Dr. Williams,

Thank you for submitting your manuscript to PLOS ONE. After careful consideration, we feel that it has merit but does not fully meet PLOS ONE’s publication criteria as it currently stands. Therefore, we invite you to submit a revised version of the manuscript that addresses the points raised during the review process.

I have received comments on your revised manuscript from two reviewers, one of whom also reviewed your initial submission. Both reviewers agreed, as do I, that your revised manuscript is an improvement on the initial version, but some additional edits are required, primarily for clarity. Please go through their comments and respond to them, as well as a handful of additional comments from me (see below). Note that one reviewer's comments are contained in a Word document that you should be able to access (in PLOS ONE Editorial Manager if it's not attached here). Also, you will need to indicate how readers can access the data from the study, which you stated are fully available without restriction. Per PLOS ONE policy, the data should be provided as part of the manuscript or its supporting information or deposited to a public repository.

We look forward to receiving your revised manuscript.

Kind regards,

Frank H. Koch, PhD

Academic Editor

PLOS ONE

Journal Requirements:

Additional Editor Comments:

Line 49 - delete comma after "forests" (also noted by a reviewer)

Line 185 - delete "is plausible" (also noted by a reviewer)

Line 186 - replace "his" with "this"

Line 194 - "i" should be lower-case (also noted by a reviewer)

Lines 233-236 - this sentence could form part of your response to the reviewer's question (referencing line 86) about why you focused on 12 particular bird species in your study.

Line 260 - replace "or" with "of"

Lines 343-345 - This sentence reads a little awkwardly. I suggest inserting "whether" after "explore"

Line 417 - "autecological" seems like an odd word choice here. Why not just say "ecological needs"? (also noted by a reviewer)

Line 419 - delete extra space after "[71]"

Line 431 - delete extra space after "met"

Line 452 - "2020 NJ Forest Action Plan" - reading this again reminded me about a nagging question that I meant to pose during the previous review round. Your study deals with observations dating from 2017 or earlier. Has anything changed in the time since? It would make sense, of course, for the 2020 NJ Forest Action Plan to be based on observations and analyses from the previous decade. However, we're now seven years past the end of data collection for your study. Perhaps forest conditions, etc., haven't changed significantly in the NJPB over this time. On the other hand, it's also been a few years since the release of the Forest Action Plan. Can you comment on anything regarding implementation of the Forest Action Plan, or on any other aspects of the NJPB that may be different from several years ago?

Reviewers' comments:

Reviewer's Responses to Questions

**Comments to the Author**

1. If the authors have adequately addressed your comments raised in a previous round of review and you feel that this manuscript is now acceptable for publication, you may indicate that here to bypass the “Comments to the Author” section, enter your conflict of interest statement in the “Confidential to Editor” section, and submit your "Accept" recommendation.

Reviewer #2: (No Response)

Reviewer #3: (No Response)

2. Is the manuscript technically sound, and do the data support the conclusions?

Reviewer #2: Yes

Reviewer #3: Yes

3. Has the statistical analysis been performed appropriately and rigorously? 

Reviewer #2: Yes

Reviewer #3: Yes

4. Have the authors made all data underlying the findings in their manuscript fully available?

Reviewer #2: Yes

Reviewer #3: No

5. Is the manuscript presented in an intelligible fashion and written in standard English?

Reviewer #2: Yes

Reviewer #3: Yes

6. Review Comments to the Author

Reviewer #2: The authors have worked through the suggested comments implementing many of the suggested revisions. As a result, elements of the manuscript have been improved. However, in addressing some of the suggested revisions the authors have provided text that should be reviewed for clarity. Comments to this, as well as other remaining issues, appear below.

Line 191: What is the "ecological selection process"? What defines "habitat quality"? Please clarify this text.

Line 213: Consider dropping "is plausible" as you are not making the argument for plausibility here only that you included these as covariates.

Line 215: "his" should be 'this'

Lines 213-219: This is an exceptionally long, and meandering sentence. I would consider breaking this sentence into several, each with a unique topic. As is, it is really difficult to understand what you are trying to convey in this sentence.

Line 267: Conducting surveys on a roadway seems an illogical condition for not requiring IACUC, or related, approval. Perhaps this is the case but I think this would be questioned by a reader who might be familiar with, or associated with an institution that has different requirements towards IACUC protocols.

Line 397: Was your goal to explore 'if'...?

Line 405: What does improving abundance and biodiversity mean? In some instances a decrease in biodiversity through loss of non-native, negatively-affecting species would be beneficial, others situations a gain in biodiversity would be best. 'Improve' is a vague term here that needs to be reformed with specificity.

Line 431: Please say with additional detail what you mean with "more refined temporal evaluations"

Literature cited: Please check references for consistent formatting. Some references have spaces before page numbers, others not. you also have some journals listed with different names (eg. "The Auk" vs. "Auk").

Table 2: Include "parameters" in last sentence of heading as covariates, parameters, and suites are all noted in this table.

Reviewer #3: These are the first revisions that I'm providing to the authors. I was not able to find information about where the data and the code to analyze it are available.

7. PLOS authors have the option to publish the peer review history of their article (what does this mean?). If published, this will include your full peer review and any attached files.

Reviewer #2: No

Reviewer #3: No

---

## [Author Response · Author response to Decision Letter 1]

11 Mar 2024

Editor Comments:

Line 49 - delete comma after "forests" (also noted by a reviewer)

Response: Done

Line 185 - delete "is plausible" (also noted by a reviewer)

Response: Done

Line 186 - replace "his" with "this"

Response: Done

Line 194 - "i" should be lower-case (also noted by a reviewer)

Response: Done

Lines 233-236 - this sentence could form part of your response to the reviewer's question (referencing line 86) about why you focused on 12 particular bird species in your study.

Response: Thank you

Line 260 - replace "or" with "of"

Response: Done

Lines 343-345 - This sentence reads a little awkwardly. I suggest inserting "whether" after "explore"

Response: Done

Line 417 - "autecological" seems like an odd word choice here. Why not just say "ecological needs"? (also noted by a reviewer)

Response: Done

Line 419 - delete extra space after "[71]"

Response: Done

Line 431 - delete extra space after "met"

Response: Done

Line 452 - "2020 NJ Forest Action Plan" - reading this again reminded me about a nagging question that I meant to pose during the previous review round. Your study deals with observations dating from 2017 or earlier. Has anything changed in the time since? It would make sense, of course, for the 2020 NJ Forest Action Plan to be based on observations and analyses from the previous decade. However, we're now seven years past the end of data collection for your study. Perhaps forest conditions, etc., haven't changed significantly in the NJPB over this time. On the other hand, it's also been a few years since the release of the Forest Action Plan. Can you comment on anything regarding implementation of the Forest Action Plan, or on any other aspects of the NJPB that may be different from several years ago?

Response: Thanks for that question. The answer is nothing has changed. Politically, the system has been very slow (decades long call for a change in management) as well as a multiyear hiccup with covid shutting down most recent efforts. So we feel confident that the data and recommendations surrounding this paper are up-to-date.

Reviewer #2

Line 191 (line 167 clean): What is the "ecological selection process"? What defines "habitat quality"? Please clarify this text.

Response: We simplified the sentence to be more clear.

Line 213 (line 185 clean): Consider dropping "is plausible" as you are not making the argument for plausibility here only that you included these as covariates.

Response: Done

Line 215 (line 186 clean): "his" should be 'this'

Response: Done

Lines 213-219 (lines 185-291 clean): This is an exceptionally long, and meandering sentence. I would consider breaking this sentence into several, each with a unique topic. As is, it is really difficult to understand what you are trying to convey in this sentence.

Response: Done

Line 267: Conducting surveys on a roadway seems an illogical condition for not requiring IACUC, or related, approval. Perhaps this is the case but I think this would be questioned by a reader who might be familiar with, or associated with an institution that has different requirements towards IACUC protocols.

Response: True and we have clarified sentence.

Line 397 (line 345 clean): Was your goal to explore 'if'...?

Response: Done

Line 405 (line 352 clean): What does improving abundance and biodiversity mean? In some instances, a decrease in biodiversity through loss of non-native, negatively-affecting species would be beneficial, others situations a gain in biodiversity would be best. 'Improve' is a vague term here that needs to be reformed with specificity.

Response: We apologize, the correct word was “increase”

Line 431 (line 376 clean): Please say with additional detail what you mean with "more refined temporal evaluations"

Response: Ultimately, we decided to delete the sentence as it was overly informative or helpful.

Literature cited: Please check references for consistent formatting. Some references have spaces before page numbers, others not. you also have some journals listed with different names (eg. "The Auk" vs. "Auk").

Response: Done

Table 2: Include "parameters" in last sentence of heading as covariates, parameters, and suites are all noted in this table.

Response: Done

Reviewer 2 (or is it 3?) comments:

Line 49: remove comma after forests

Response: Done

Line 50: give some brief examples of how/why thinning helps birds

Response: Done

Line 57: same as in line 50 – why?

Response: Done

Line 69: what does diminished state means?

Response: Removed

Line 79: why mention mammals? If so, what is their response?

Response: Clarified

Line 86: why focusing on only the highest priority species? With a community model framework, why do you choose only 12 species instead of using the whole community to borrow statistical strength from more common species? Species of high concern are generally rare or find in smaller numbers, so using common species and sharing information would make results more robust.

Response: Our focus was on the predetermined priority conservation species as defined by the U.S. Fish and Wildlife Service’s Atlantic Coast Joint Venture and Mid-Atlantic and New England Coastal Bird Conservation Region 30 Implementation Plan. Additionally, we restricted our modeling effort to the 12 priority species which did not include nomadic and aerial coursing species (e.g., raptors), which may have home ranges that are orders of magnitude larger than those species targeted here.

Line 89: In an opposite thought of the previous comment, how do you think that grouping a small number of species of different ecologies in forest groups might be influencing your results? Do you expect an oriole to have a similar response to a warbler? Justify a little the choices in line 89 and 86 would help the reader understand your study a little more.

Response: We have attempted to clarify these sentences.

Line 98: explain what community, suite and species levels are.

Response: Done.

Line 105: you can use the three terms the first time (Scrub-Shrub/Early Successional/Young Forest), but choose only one hereafter in the text

Response: Done and we chose “Scrub-Shrub”

Line 168: since the values sum to one, aren’t they highly correlated to each other?

Response: Thank you for that question. While it is possible an individual point could have correlated values of percent ground cover since they sum to one, this is less of an issue across the diversity of site point collections. This is a standard assumption for Daubenmire frame microscale habitat collection. That being said, we are sensitive to correlation and we standardized all continuous covariates, centering each on a mean of 0 and standard deviation of 1, to improve model convergence and interpretability. We used Pearson’s correlation tests to assess the degree of collinearity of model parameters and did not fit models when |r| > 0.7. This fact is later explained in the methods. And then we talk about removal of correlated variables in the results.

Line 172 and 177: did you only measure canopy cover for non-grassland sites?

Response: We collected at all site but of course the answer was always “no canopy cover” in early successional sites.

Line 190: It would be good to test and mention that your covariates are not correlated to each other

Response: We apologize, we do not understand this comment in relation to Line 190. However, we think our discussions of the correlation topics in the later paragraphs should address this comment.

Line 194: lower case i

Response: Done

Line 261: can you justify a little why you chose such a large value of correlation threshold? 0.7 is a lot. I would remove variables that are so correlated with each other.

Response: The range of correlation differs across studies but when dealing with datasets with large sample sizes (as we had) the use of 0.7 is robust to still detect biologically significant results hence our use of this value.

Line 269: you mention sampling birds at 0-50m and 50-100m radii. where was this used in the model? If not, remove from text.

Response: I’m sorry, we do not have reference to this in the text so we believe this is already done.

Line 274: why (%)? Remove; as mentioned before, including variables that have percentages make the levels within the same variable very correlated to each other, e.g. if one variable is 40%, the other will be 60%, if one is 15%, the other will be 85%. I would check the correlation between the levels of the variable that you are including in the model. I would mention the variables that you removed from the model in the methods (when you mention the 0.7 threshold) and not have it again here. Because you mention this in the results, the reader has no idea of how many parameters you are actually using. Showing the vectors with all your environmental covariates in the methods section will help.

Response: I suppose the issue is whether to include the correlation results in the Methods or the Results. We thought about this request and ultimately decided to keep it in the results. We felt it was more important to talk about the wide range of variables that we collected in the field in the Methods and then present the pairing of the data (due to correlations) for our model building as part of our analysis/results. Taking out the first 2/3rds of the first paragraph of the results to be inserted in the Methods made the presentation of the remaining sentences of the first paragraph of the Results confusing.

Line 282: r < 0.35, not greater

Response: Done

Line 287: a figure would be very helpful for the values in table 1

Response: We appreciate this recommendation, however after considering the wide range of the values, we ultimately decided it would be hard to see the differences in the box and whisker plot and decided to stick with the specificity of numbers in a table. We apologize.

Line 288: I would mention the binning in the methods section, or not even mention it at all and just have the 0-100m radius in the text.

Response: I’m sorry, there is no mention of 0-50 and 51-100 binning in the current version of the text so we believe this has been resolved.

Line 292: include the traceplots and a table with the r-hat values.

Response: We have added a table of the correlations to supplement the first paragraph of results.

Line 304: it is still very unclear what community is – is it all the 3 suites combined? If so, does it make sense to clump species with such different ecology and habitat choices, as well as opposite responses to vegetation variables in a group? You should clearly define what the bcr, community and suite levels are.

Response: On line 30 and 101 we explain what community at the Bird Conservation Region 30 and suite habitats are. While we primarily interested in the suite level habitats since birds are evolved to those, we still felt it was important to look at the pine barrens as a whole to address ecosystem habitat management impacts of the birds of conservation concern. 

Line 305: show the histogram of the posteriors for the parameters. You can put both these and the traceplots in supplementary. Posterior predictive checks and pairs plots should be in supplementary too.

Response: We have included a supplemental table of pairwise correlations of habitat variables but decided to keep Figure 2 and Figure 3 in the main body of the text. 

Line 308: because of the nature of your standardized coefficients, it is not clear what does a increase of 0.46 birds is. Can you put it in a scale of 1 or 10 birds for each x amount of basal area

Response: Done

Line 343: since you focused on high priority species, maybe you should say that your goal was to ensure their presence, and not the presence of the whole community. A more diverse community would include all species, and not focus on a subset.

Response: Done

Line 343: how does that result applies to grasslands? Could you please have a table with all the covariates used for each of the three suites of habitats? Or did you just put zero for the basal area in grasslands? Explain how you made the choice of putting a zero or removing the variable for one of the suite levels. Also, please explain in the methods how variables were measured for grasslands. You only have a species for grasslands. Given that, is it worth it to include it?

Response: There are no true grasslands prairies in the New Jersey Pine Barrens but we can have grassland dominant species occurring in open managed pine savannah settings. Therefore, all microhabitat measurements are plausible with no guaranteed zeros. 

Line 349: but did the early successional and grassland species increased in the reduced basal area pine stands, or in the early successional and grasslands that have reduced basal area? If it was not in the pine stands, the diversity didn’t really increase with the thinning, only in the whole NJPB. I think you meant that the diversity increased in the grasslands and early successional with thinning on those areas – could you make it a little clearer in the text. Also mention in what habitat these results are true for in the 9 citation.

Response: This relates to the last questions and response. There are no grasslands in the pine barrens and so we are discussing reducing basal density of the pine savannah system. For these comments we added a few words to hopefully make it clearer.

Line 363: why did species had such different responses? Were the group responses similar? 

Response: This is just a function of certain variations that occur in niche portioning across species even if general preference for certain types of ecosystems.

 We have added a clarification. 

Line 417: autecological?

Response: Fixed

Line 446: are those recommendations for forest? Grasslands? Early succession? And please explain again where that number came from.

Response: As mentioned before there are no grasslands and this is just a pine forest or pine-savannah system but we have clarified the sentence. 

Line 448: same as before – include a citation for that recommendation, or briefly mention what result that you have that indicate this statement.

Response: Done

Figure 2: you can make the x-axis smaller (like panel 1 from -0.25 to 0.3) to highlight your results better and make the values clearer – they look a little small. I would also write the variable names on the axis titles and put the symbol from the equation (beta_grass) in parenthesis. I would also put the units or mention they were standardized.

Response: We are terribly sorry but our statistician who made this graph had a personal emergency since the first submission of the manuscript and left the project with no forwarding contact information. We are now not able to make a new graph given the limited files he left us with.

General: 

I would focus more on why the forest covariates that you choose are relevant for birds, and how they affect the different birds in the different habitat types. I understand that the NJPB is an important and super relevant conservation area for birds, but I would focus a little more in why the covariates are helping (or not) bird populations, rather than so much in how the distribution of those variables in NJPB is. I missed in the discussion some more ecological explanations why the 3 different groups of species respond the way they do to thinning – for example, how would a more open understory benefit a grassland species (food availability, nest site availability, etc), but that was not true for the forest species. This is super interesting research about different treatment types, but also about how and why species respond to them.

Response: With the greatest of respect, we feel like our existing discussions does in fact provide numerous examples that discuss this topic, however we have added a few clarifying sentences throughout.

I would also explain well how you measured the different variables for the different habitat types (like basal area for forest versus grasslands), and if it makes sense to have a BCR response that includes these two types of responses, since they can have opposite signals within the groups, making the BCR estimate zero.

Response: As mentioned previously, we are working in a pine and pine sava

---

## [Editor Report · Decision Letter 2]

28 Mar 2024

Active forest stewardship benefits priority birds in the New Jersey Pine Barrens

PONE-D-23-17857R2

Dear Dr. Williams,

We’re pleased to inform you that your manuscript has been judged scientifically suitable for publication and will be formally accepted for publication once it meets all outstanding technical requirements.

Kind regards,

Frank H. Koch, PhD

Academic Editor

PLOS ONE

Additional Editor Comments (optional):

Thank you for completing another set of revisions. I think that you have addressed all of the reviewer comments sufficiently. I noticed a couple of very minor things in the text that you may want to change, which I've noted below. Regarding the comments from Reviewer 3 (they called themselves Reviewer 2) about the binning of detection distances, you do mention this in lines 152-153 of the current revision, but I consider this issue resolved based on what you said in lines 289-292, i.e., that ultimately you pooled across the distance class bins. In any case, I believe your manuscript is now suitable for publication.

Line 71 - I suggest "cycles" instead of "cycle".

Line 253 - maybe "the observational process" or "observational processes".

Supplemental Material Table 1 - "forbs" instead of "forbes".
---

## [Editor Report · Acceptance letter]

25 May 2024

PONE-D-23-17857R2 

PLOS ONE

Dear Dr. Williams, 

I'm pleased to inform you that your manuscript has been deemed suitable for publication in PLOS ONE. Congratulations! Your manuscript is now being handed over to our production team.

Kind regards, 

on behalf of

Dr. Frank H. Koch 

Academic Editor

PLOS ONE